# Behavioral interplay between mosquito and mycolactone produced by *Mycobacterium ulcerans* and bacterial gene expression induced by mosquito proximity

**Dongmin Kim** [1¤]*, **Tawni L. Crippen**[2], **Laxmi Dhungel**[3], **Pablo J. Delclos**[4], **Jeffery K. Tomberlin**[1]*, **Heather R. Jordan**[3]*

**1** Department of Entomology, Texas A&M University, College Station, Texas, United States of America, **2** Southern Plains Agricultural Research Center, Agricultural Research Service, USDA, College Station, Texas, United States of America, **3** Department of Biological Sciences, Mississippi State University, Starkville, Mississippi, United States of America, **4** Department of Natural Sciences, University of Houston-Downtown, Houston, Texas, United States of America

¤ Current address: Florida Medical Entomology Laboratory, University of Florida, Vero Beach, Florida, United States of America

* dongminkimkorea@gmail.com (DK); jktomberlin@tamu.edu (JKT); jordan@biology.msstate.edu (HRJ)

**Data Availability Statement:** All relevant data are within the paper and its Supporting Information files. The RNASeq data that support the findings of

## Abstract

Mycolactone is a cytotoxic lipid metabolite produced by *Mycobacterium ulcerans*, the environmental pathogen responsible for Buruli ulcer, a neglected tropical disease. *Mycobacterium ulcerans* is prevalent in West Africa, particularly found in lentic environments, where mosquitoes also occur. Researchers hypothesize mosquitoes could serve as a transmission mechanism resulting in infection by *M. ulcerans* when mosquitoes pierce skin contaminated with *M. ulcerans*. The interplay between the pathogen, mycolactone, and mosquito is only just beginning to be explored. A triple-choice assay was conducted to determine the host-seeking preference of *Aedes aegypti* between *M. ulcerans* wildtype (MU, mycolactone active) and mutant (MU$^{lac-}$, mycolactone inactive). Both qualitative and quantitative differences in volatile organic compounds' (VOCs) profiles of MU and MU$^{lac-}$ were determined by GC-MS. Additionally, we evaluated the interplay between *Ae. aegypti* proximity and *M. ulcerans* mRNA expression. The results showed that mosquito attraction was significantly greater (126.0%) to an artificial host treated with MU than MU$^{lac-}$. We found that MU and MU$^{lac}$ produced differential profiles of VOCs associated with a wide range of biological importance from quorum sensing (QS) to human odor components. RT-qPCR assays showed that mycolactone upregulation was 24-fold greater for MU exposed to *Ae. aegypti* in direct proximity. Transcriptome data indicated significant induction of ten chromosomal genes of MU involved in stress responses and membrane protein, compared to MU$^{lac-}$ when directly having access to or in near mosquito proximity. Our study provides evidence of possible interkingdom interactions between unicellular and multicellular species that MU present on human skin is capable of interreacting with unrelated species (i.e., mosquitoes), altering its gene expression when mosquitoes are in direct contact or proximity, potentially impacting the production of its VOCs, and consequently leading to the stronger attraction of

this study are openly available in NIH Bioproject with accession number PRJNA729466 (https://www.ncbi.nlm.nih.gov/bioproject/PRJNA729466).

**Funding:** This work was partially supported by the Texas A&M AgriLife Research Insect Vector Disease Program #505320-90360 (J.K.T. and D.M.K). The funders had no role in study design, data collection and analysis, decision to publish, or preparation of the manuscript.

**Competing interests:** The authors declare that the research was conducted in the absence of any commercial or financial relationships that could be construed as a potential conflict of interest.

mosquitoes toward human hosts. This study elucidates interkingdom interactions between viable *M. ulcerans* bacteria and *Ae. aegypti* mosquitoes, which rarely have been explored in the past. Our finding opens new doors for future research in terms of disease ecology, prevalence, and pathogen dispersal outside of the *M. ulcerans* system.

## Introduction

While public interest regarding newly emerging infectious diseases, such as SARS-CoV-2, is becoming a center of global research, other important tropical diseases remain neglected. In many cases, mosquitoes are the mechanism allowing for the transport, or transmittal, of the causative agents of these tropical diseases [1]. One such disease, Buruli ulcer (BU), is prevalent in tropical and subtropical regions [2, 3] and is considered endemic in West African countries [4–7]. *Mycobacterium ulcerans* (MU) is the etiological agent of BU and is commonly colonized in aquatic environments including lentic water or vegetation [8, 9]. Unlike other *Mycobacterium* species, MU produces mycolactone (molecular formula: $C_{44}H_{70}O_9$) that is a cytotoxic lipid toxin. Mycolactone is composed of a 12 membered macrolide ring structure and two side chains (S1 Fig) [10–12]. Mycolactone is synthesized by polyketide synthases encoded by the extrachromosomal plasmid pMUM001 and can diffuse through plasma membranes [10, 13]. Several molecular targets of mycolactone have been identified (S1 Fig). For instance, mycolactone directly binds to Wiskott–Aldrich syndrome protein (WASP) and Sec61 thereby preventing cytoskeleton formation and co-translational translocation of proteins, respectively [14–22]. Further, mycolactone causes immunomodulation by preventing maturation, migration, and agranulocytes-chemoattractant production in dendritic cells [23]. Mycolactone-mediated cytopathicity and apoptosis of macrophages and dendritic cells have been observed [24]. Additionally, mycolactone binds to angiotensin II (A 375 T2) receptors to hyperpolarize neurons [25]. For an excellent review, please see Guenin-Macé et al. [26] and references therein. Due to these mechanisms, BU disease has novel clinical symptoms that consist of a painless skin ulcer, tissue necrosis, bone deformation, and possible delayed wound healing and secondary infections that can in severe cases lead to death [8, 27]. The mode of BU transmission to humans remains unclear but previous studies showed that human activities through water contact (e.g., bathing and fishing) or insect bites may have been a risk factor for BU infection [27, 28].

Recent evidence indicates mosquitoes could also serve in the mechanical transmission of MU [29–31]. However, the exact relationship between MU and mosquitoes has not been truly elucidated. From an epidemiological and future control standpoint, deciphering the mechanism of possible interplay between mosquitoes and MU is critical. If such a mechanistic interaction is true, it could be harnessed and developed (i.e., manipulate vector-pathogen interaction) into a means to suppress MU transmission and resulting BU in a given community.

Such interactions are well-grounded in research identifying cues used by mosquitoes to locate hosts or oviposition sites and thereby provide credence for the current research. A number of environmental cues are used by mosquitoes to locate hosts. In addition to vision [32], abiotic factors, such as carbon dioxide [33], and heat [34], play a role; however, mosquitoes are known to heavily rely on olfactory sensory discriminating chemical cues associated with hosts [35]. As determined previously, microbes associated with human skin convert odorless human skin residues (i.e., sweat) to aliphatic and aromatic carboxylic acids [36, 37]. These volatile organic compounds (VOCs) are the principal cues used by female mosquitoes for host blood acquisition [35, 38].

A potential cue for attracting mosquitoes to individuals with MU present on their skin is the mycolactone [39]. Sanders et al. [40] demonstrated artificial blood-feeders coated with mycolactone attracted significantly more yellow fever mosquitoes, *Aedes aegypti aegypti* (L.) (Diptera: Culicidae) than control blood-feeders [8, 9, 41]. Additionally, the mosquito response was dose-dependent with a 29.2% greater attraction to the blood-feeder treated with a high mycolactone dose (1.0 µg/mL) than a low dose (0.05 µg/mL) feeder. This discovery is intriguing because this mosquito species commonly occurs in areas endemic to BU disease and thus, they share the environment, such as lentic aquatic habitats [8, 42] or the vegetation [9] in standing water where MU occurs.

This relationship extends beyond attraction to a blood-meal as mosquitoes show a preference for oviposition in areas containing mycolactone [40]. Mashlawi et al. [43] demonstrated mosquitoes reared in water with mycolactone were more likely to oviposit in similar habitats. However, in both studies, the researchers were working directly with mycolactone extracted from populations of MU, rather than bacterial cells. Furthermore, despite these findings, the mechanistic responses of mosquitoes beyond behavioral phenotypes have not been elucidated.

In this study, we determined the response of female *Ae. aegypti* mosquitoes to wildtype MU (mycolactone active) and mycolactone mutant (mycolactone inactive), MU$^{lac-}$, through a behavioral study utilizing artificial blood-feeders. The VOC profiles from MU and MU$^{lac-}$ were also measured to give insight into specific compounds cueing attraction by mosquitos to this bacterium. Additionally, we asked whether mosquitos themselves influence MU gene expression by measuring MU and MU$^{lac-}$ mRNA expression when mosquitos were directly in contact or only in near proximity compared to when no mosquitoes were present. Results from this study provide fundamental explanations for mosquito-MU interactions and potentially peel away another layer from the complex BU epidemiological phenomenon.

## Materials and methods

### Mosquito colony

*Aedes aegypti aegypti* (Liverpool strain) were maintained with a standardized mosquito-rearing schedule [44] in a colony held in an environmental chamber (25.0 ± 0.5˚C, 65.0 ± 5.0% RH, and a photoperiod of 12:12 (L:D) h). at the Forensic Laboratory for Investigative Entomological Sciences (F.L.I.E.S. Facility) at Texas A&M University, College Station, Texas, USA.

### Bacteria

Wildtype *M. ulcerans* 1615 (MU) and mycolactone mutant *M. ulcerans* 1615::TN118 (MU$^{lac-}$) [45] were used for the experiments. MU$^{lac-}$ produces neither the core nor the side chain of mycolactone due to an insertion in mup045 and is easily visualized by the lack of pigmentation [39, 46]. Both were grown on Middlebrook (MB) 7H10 agar (Difco Labs, USA) supplemented with 10% (v/v) oleic acid, albumin, dextrose supplement (OADC) with or without hygromycin (50 µg/mL; Sigma-Aldrich Corp., USA) at 32˚C for 6 to 8 weeks. Four sterilized 25 mm (dia.) filter disks (Whatman, UK) were placed on top of the agar prior to complete solidification in a 100mm bacteriological plate and 100µl of MU or MU$^{lac-}$ in PBS at an optical density (OD$_{600}$) ≈ 1.2 was spread directly onto the plate. After 6–8 weeks, the filter disks were aseptically removed from the plate and used for the experiment (Fig 1A and 1B).

### Blood feeders

Blood-feeders were individually constructed of a 25-mL sterile tissue culture flask (Corning Inc., USA) tightly wrapped with parafilm and secured with cellophane tape. A 1 mL aliquot of

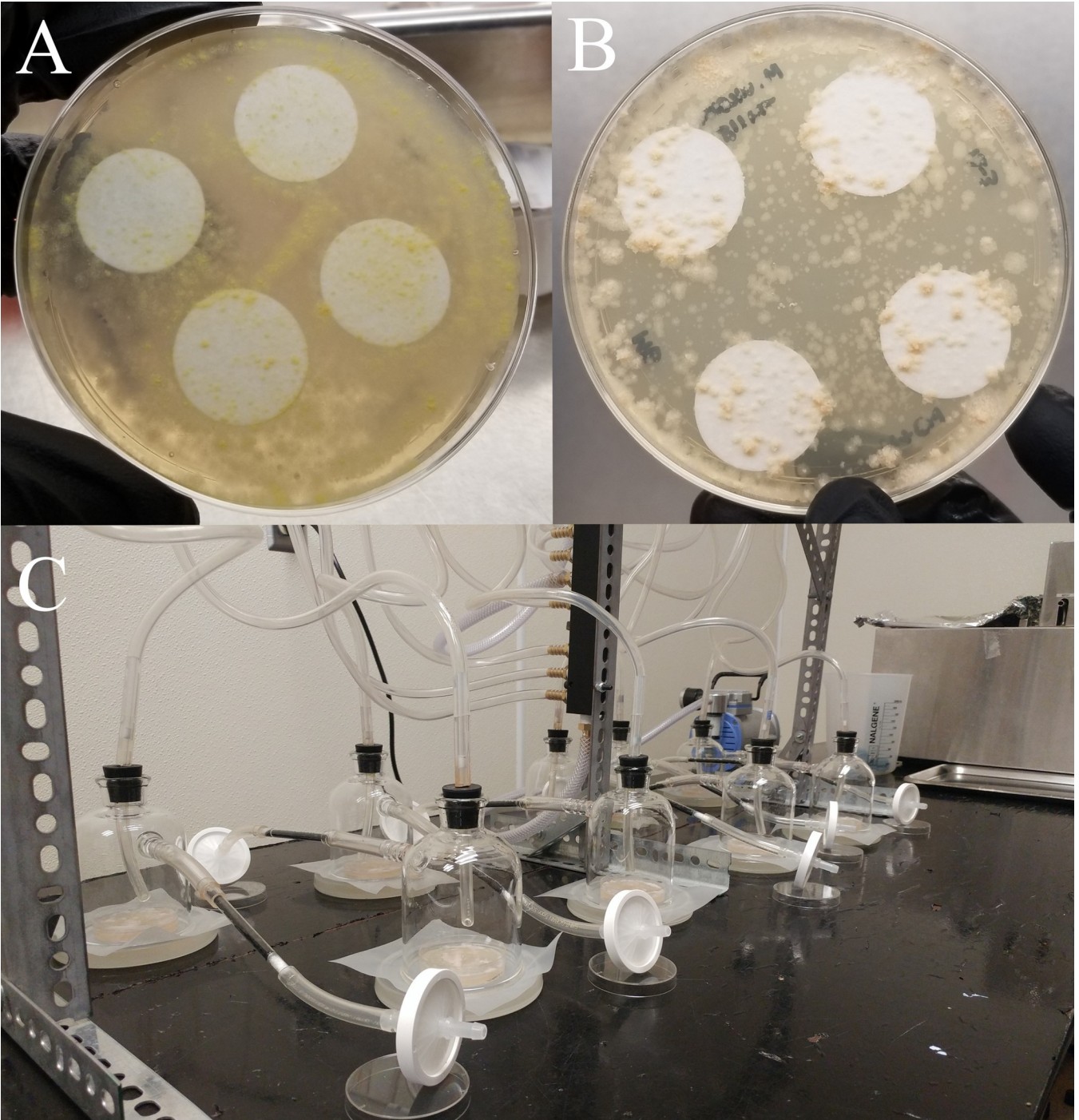

**Fig 1.** Morphological features of *Mycobacterium ulcerans*; (A) MU (*M. ulcerans*) wildtype: mycolactone active (B) MU-mutant: mycolactone inactive (C) Closed-loop-stripping apparatus for VOC collection.

defibrinated rabbit blood (HemoStat Laboratories, USA) was injected into the space between the culture flask and parafilm. A 5.0 x 5.0 cm autoclaved piece of 100% cotton gauze (Dynarex Co., USA) was placed over the parafilm. A filter disk grown with MU, MU$^{lac-}$, or sterile PBS as a control was inserted between the cotton gauze layer and the parafilm.

## Mosquito behavior assay

Experiments were a modification of methods previously described [47]. Host-seeking preference of *Ae. aegypti* by comparison between MU, MU[lac-], and PBS inoculated filter disk was determined by aggregation behavior of *Ae. aegypti* to blood-feeders treated with *M. ulcerans* wildtype or mutant inoculated filter. Briefly, 2 h prior to each trial, 50 mated female mosquitoes (3–5 d old) that had never received a blood meal, were collected using an aspirator (Hausherrs Machine Works Co., USA). Mosquitoes were released into a clear Plexiglas™ cage (82 x 52 x 45 cm) with an aluminum wire mesh top and allowed to acclimate at 24.0 ± 1.0°C, relative humidity of 65.0 ± 5.0%. Experiments were performed 30 min after sunrise (chamber at 12:12 L:D), which corresponds to the normal biting activity of *Ae. aegypti* [48]. Three blood-feeders, each with a separate treatment (MU, MU[lac-], or PBS), were connected to a water bath (Thermo Fisher Scientific, USA) and maintained at 37°C, were placed in parallel, 16.5 cm apart, gauze side down on the wire mesh top [40].

For each experiment, three trials were performed in succession by rotating each of the three treatments to each of the three different locations initially assigned by a random number generator and rotated clockwise across trials to prevent positional bias. All equipment was cleaned with 3% Lysol and then 95% ethanol and allowed to ventilate between trials. During the experiments, mosquito landing activity at each blood-feeder was recorded with a camera (2160p / 30fps; LG, Korea) mounted on the outside of the cage. The total number of mosquitoes responding by landing and touching each blood-feeder (response) was determined for each minute over a 15 min assay period. Each experiment was replicated three times (9 trials total).

## Statistical analysis for mosquito behavior assay

The odds ratios for selecting a particular treatment by a blood-feeder were analyzed and visualized using R version 3.4.3 and the DescTools package. Normality of the data was assessed using the Shapiro-Wilk test in JMP® statistical software version 13 (SAS Institute Inc., USA). To determine if there were significant differences in mosquito response rates among treatments and across nine trials, analysis of covariance (ANCOVA) was performed. A generalized linear mixed model (GLMM) was used to compare mosquito responses across treatments. The significance of differences in the probability (P) of response (attraction) by *Ae. aegypti* to the different treatments was assessed with a significance level of $p \leq 0.05$.

## VOCs collection assay

Each suspension including 100μl of MU, MU[lac-] in PBS at an optical density ($OD_{600}$) = 1.2 or PBS as a control was inoculated into four sterilized 25 mm (dia.) filter disks on top of the MB agar. After 6–8 weeks, the filter disks were aseptically removed from the MB agar and used for VOC collection. Bacterial volatiles were collected by the closed-loop-stripping-analysis (CLSA) technique at 24.0 ± 1.0°C, relative humidity of 65.0 ± 5.0% (Fig 1C) and analyzed in nonuplicate trials of quadruplicate replicates (n = 36) from the following samples: 1) MU inoculated filter disk collected from MB agar 2) MU[lac-] inoculated filter disk collected from MB agar with 50 μg/mL hygromycin (mycolactone production inactivated by insertion of a hygromycin-resistance gene) 3) PBS as a MU control, inoculated filter disk collected from MB agar 4) PBS as a MU[lac-] control, inoculated filter disk collected from MB agar with 50 μg/mL hygromycin. Two controls were used to eliminate the redundancy of any vaporized chemicals from an MB growth medium. The protocol was modified from the previous study [49] and designed to improve filtration in the incoming air. Before every headspace sampling from bacteria, the apparatus was thoroughly cleaned with dichloromethane ($CH_2Cl_2$) and autoclaved at 121°C for 15 min.

Each filter disk was transferred to a 7.5 x 11 cm (O.D x H) glass filtering jar (Kimble Chase, USA) with a ground flat glass and sealed with a parafilm. The rubber stopper on the top of the glass filtering jar was equipped with one hole and inserted with a volatile trap packed with approximately 30.0 mg of Hayesep® Q porous polymer (Volatile Assay Systems, USA) connecting vacuum pump (Rocker, Scientific Co., Ltd., Taiwan) with Tygon® tubing (Saint-Gobain S.A., USA). The tooled hose connected with 3 cm of Tygon® tubing piece inserted with a 0.2 μm in pore size of the bacterial filter (Midwest Supplies, USA) and a 14.6 cm carbon filtered pipet (Marineland, Cincinnati, Ohio) to purify incoming air. Samples from each MU, $MU^{lac-}$, and PBS were obtained by running the apparatus at 1 L $min^{-1}$ for 1 h, respectively.

Gas chromatography–mass spectrometry (GC-MS) analyses were carried out on Agilent 6890 gas chromatograph with an Agilent 5973 mass selective detector (Agilent Technologies, USA) at Environmental Research Group at Texas A&M University in College Station, Texas. Conditions are as follows: 1.29 mL $min^{-1}$, injection volume: 1μL; transfer line: 300°C. The GC was programmed as follows: 8 min at 35°C, increasing at 6°C $min^{-1}$ to 160°C and operated in split mode: 60 s at 250°C. The carrier gas was Helium at 1 mL / $min^{-1}$. Candidate identification of compounds was made by matching the comparison of mass spectra with the mass spectra fragmentation patterns in the NIST05 mass spectra library for the peak observed in the chromatograms.

## Statistical analysis for VOCs

The GC-MS data were processed to determine the percentage of each compound's area in each sample, including the control group. To assess volatile profile differences, PERMANOVA was performed using the Adonis function in R (version 3.4.3) with the vegan package. The VOC profiles were analyzed using NMDS with the Bray-Curtis distance matrix, reducing the data into a two-dimensional space. Indicator species analysis was conducted to identify influential compounds for each group, with a stress value threshold of <0.2. Compound abundance was compared using a two-way ANOVA with Tukey-Kramer HSD post hoc test in JMP® statistical software version 13 (SAS Institute Inc., USA) with a significance level of $p \leq 0.05$.

## Mosquito proximity assay

The behavior assay design described previously was also used for three proximity (i.e., distance between mosquitoes and bacteria) levels: 1) Direct—50 mosquitoes in the Plexiglas box in direct contact with the bacteria via blood-feeders placed directly onto the aluminum wire mesh top; 2) Near—50 mosquitoes in the Plexiglas box in close proximity via blood-feeders placed 3 mm above the aluminum wire mesh top using sterile glass hematology slides supporting two opposing edges and thus elevating the feeders; and 3) None—no mosquitoes in the Plexiglas box with the bacteria via blood-feeders placed directly onto the aluminum wire mesh top. After 15 min of exposure, filter disks with bacteria were immediately placed into cryovials and frozen at -80°C until RNA isolation and transcriptome analyses.

In a separate experiment, MU ($10^6$ CFU/mL) was exposed in opened tubes to newly emerged, adult *Ae. aegypti* mosquitos at three different distances for 24 h: Direct (0.025 m, N = 6), Near (1.83 m, N = 3), and Far/None (30.5 m, N = 6). Immediately after exposure, samples were spun down and RNA was isolated using the methods below. RNA was converted to cDNA (with appropriate controls) using the Verso cDNA Synthesis Kit according to the manufacturer's instructions. Briefly, the cDNA reaction mixture (4μl synthesis buffer, 2μl dNTP mix, 1μl random hexamer, 1μl Verso enzyme, and 1μl RT enhancer and the template) was heated at 42°C for 1 h to obtain cDNA. cDNA was used to measure gene expression of the enoyl reductase (ER) gene, a gene on the plasmid for mycolactone production, using *ppk* as a

housekeeping control for normalization [41]. The RT-qPCR data was analyzed using the ΔΔCT method to determine fold change relative to housekeeping control (*ppk*) and significant difference (p<0.05). Resulting regulation of individual samples was determined relative to the mean of control (Far/None) samples, which was considered baseline. A fold change greater than 1 was considered as upregulated. In an event the fold change fell between 0 and 1, the negative of the reciprocal of fold change was calculated to determine downregulation. The experiment was performed two times.

## Transcriptome analysis

RNA was isolated from MU and MU<sup>lac-</sup> using TRIzol™ Reagent (Thermo Fisher, USA) and standard protocol. Briefly, individual frozen filters were added to 1 mL of TRIzol™ Reagent and 0.1 mm glass beads. Samples were homogenized in a bead beater with phase separation following chloroform addition. RNA was precipitated with isopropanol, washed with 75% ethanol, resuspended in 50 μL of nuclease-free water, and incubated at 60°C for 15 min. Samples were purified using the PowerClean® Pro RNA Clean-Up Kit (MO BIO Laboratories, Inc., USA), and treated with Turbo DNAse (Thermo Fisher, USA) according to the manufacturer's instructions, as necessary. RNA was quantified using a Qubit 2.0™ (Life Technologies, USA), integrity determined through gel electrophoresis, and stored at -80°C until further processing for RT-qPCR or library preparation. RNA libraries were created using the NEBNext® Ultra™ RNA Library Prep Kit and NEBNext® Multiplex Oligos (Dual Index Primers) for Illumina® (New England BioLabs, USA) using protocols for purified mRNA or rRNA depleted RNA. Quality control and high-throughput sequencing were performed by St. Jude Children's Research Hospital on an Illumina HiSeq2000 with 2 X 150 bp paired-end read lengths. Sequences were initially trimmed using TrimGlare v0.4.2 [50], then by a more stringent quality trimming using default parameters within the Qiagen CLC Workbench 20.4.1 (https://www.qiagenbioinformatics.com/). Resulting high-quality reads were aligned to the MU *Agy99* (NC_008611.1) and plasmid pMUM001 complete sequences (NC_005916.1) using CLC Genomics Workbench 20.4.1 (Qiagen, Germany). Sequence data were archived in the NCBI SRA database (submission number SUB9616294 and BioProject PRJNA729466). RNASeq data were mapped with the following parameters: (a) maximum number of allowed mismatches set at 2, with insertions and deletions set at 3; (b) Length and similarity fractions were set to 0.8, with autodetection for both strands; (c) minimum number of hits per read was set to 10. Gene expression values were reported as normalized reads per kilobase of transcript per million (RPKM) mapped reads. Per sample normalization was performed using the TMM (trimmed mean of M values) method of [51], followed by the TMM-adjusted log CPM (counts per million) counts, and cross-sample Z-Score normalization. Reads with FDR-adjusted p-value less than or equal to 0.05 were considered significant. Transcripts were further annotated into pathways by linking protein ID with potential conserved domains and protein classifications archived within the Conserved Domain Database [52], and using KEGG and STRING databases [53, 54], and NC_08611.1 and NC_005916.1 annotation. A Principal Component Analysis plot was created to visualize differences between normalized MU and MU<sup>lac-</sup> transcripts according to mosquito proximity.

## Statistical analysis for transcriptome analyses

Differential expression of TMM normalized reads was measured using multi-factorial statistics based on a negative binomial Generalized Linear Model. MU chromosome transcripts were compared using the Likelihood Ratio or Wald Test against MU<sup>lac-</sup> according to strain while controlling for proximity. Individual chromosome and plasmid genes found significant with

the former were further analyzed between individual treatment conditions, according to strain and proximity using ANOVA with Bonferroni correction. $Log_2Fold$ change and adjusted p values were reported.

## Results

### Mosquito behavior

Mosquito responses between MU, MU^lac-, and Phosphate-buffered saline (PBS) as a control were significantly different. In relation to the control, odds ratio analysis indicated mosquito response to the blood-feeder treated with MU was significantly greater, 4.77 (p<0.0001), whereas the response to MU^lac- was 1.25 greater (Fig 2A). A significant interaction among treatment types (MU, MU^lac-, and PBS) was detected over time ($F_2$ = 288.1908; p<0.0001). No significant trial and time effect interactions ($F_1$ = 1.9856; p = 0.1596) were found.

The mean number of mosquito responses to blood-feeders with different treatments was compared over each time point (min) for the nine trials (Fig 2B). Both MU and MU^lac- treatments elicited the lowest number of responses by mosquitoes to the blood-feeders at 1 min, 15.7 ± 3.6 and 9.2 ± 2.1, respectively (Fig 2C). The PBS blood-feeder elicited the least mosquito response at all the time points beginning with the lowest response, 7.7 ± 1.6, at 1 min. Responses to MU were significantly higher than MU^lac- at 2 (97.0%), 3 (104.8%), 4 (103.4%), 5 (105.8%), 6 (112.2%), 7 (146.3%), 8 (137.4%), 9 (137.2%), 10 (156.3%), 11 (130.3%), 12 (133.1%), 13 (159.3%), 14 (122.0%), and 15 (148.7%) min post-exposure. The peak mean number (± SE) of mosquitoes responding to MU, MU^lac-, and PBS occurred at 10 min (36.4 ± 3.3), 11 min (14.7 ± 1.7), and 10 min (13.3 ± 2.4), respectively (Fig 2B).

The total number of attraction responses (7,207) from the nine trials over the 15 min experimental period to MU, MU^lac-, and PBS treatments were 3,980 (55.2%), 1,761 (24.4%), and 1,466 (20.3%), respectively (Fig 2D). The total number of MU responses was significantly higher than the other treatments ($F_2$ = 234.3004; p<0.0001); 126.0% and 171.5% higher than the MU^lac- and PBS, respectively. Total mosquito responses to MU^lac- were not significantly different (p = 0.0537) than to PBS (Fig 2D). As a solvent control experiment, no bias in mosquito responses between a blood-feeder with or without PBS was confirmed [55].

### Bacterial VOC composition

Twenty nine compounds were identified from the headspace volatiles among samples by comparing the experimental mass spectra with the NIST14 Mass Spectral Library (Table 1 and S2 Fig). MU had a mean of 11.78 ± 0.49 compounds compared to 11.56 ± 0.50 with MU^lac-. The mean of compounds detected from each control: 1) PBS inoculated filter disk from MB agar and 2) PBS inoculated filter disk from MB agar with 50 μg/mL hygromycin, was 11.00 ± 0.29 and 11.33 ± 0.41, respectively. Excluding octane, which was added as an internal standard, 20 compounds were identified from MU, 17 of which were also common to MU^lac-. Nine VOCs were shared by all treatments: benzaldehyde, phenol, nonanal, benzothiazole, nonadecane, tetradecane 2,6,10-trimethyl, eicosane, butylated hydroxytoluene, and diethyl phthalate (Table 1). After all compounds shared by treatments including controls were excluded, six compounds were unique to MU based on the relative frequency and abundance across treatments; styrene, phenylamino, (trimethylsilyl) methyllithium, naphthalene, heneicosane, and methyl stearate; and one compound was unique to MU^lac-; 1Butanol,3-methyl (Table 2 and S3 Fig). Relative abundance and quantity range percentages, and biological relevance are summarized in Table 2, organizing VOCs by retention times. Differential VOC profiles between MU and MU^lac- were statistically determined by ANOSIM (R = 0.5751, p = 0.003). The stress value representing the accuracy in spatial similarity/dissimilarity was 0.1257.

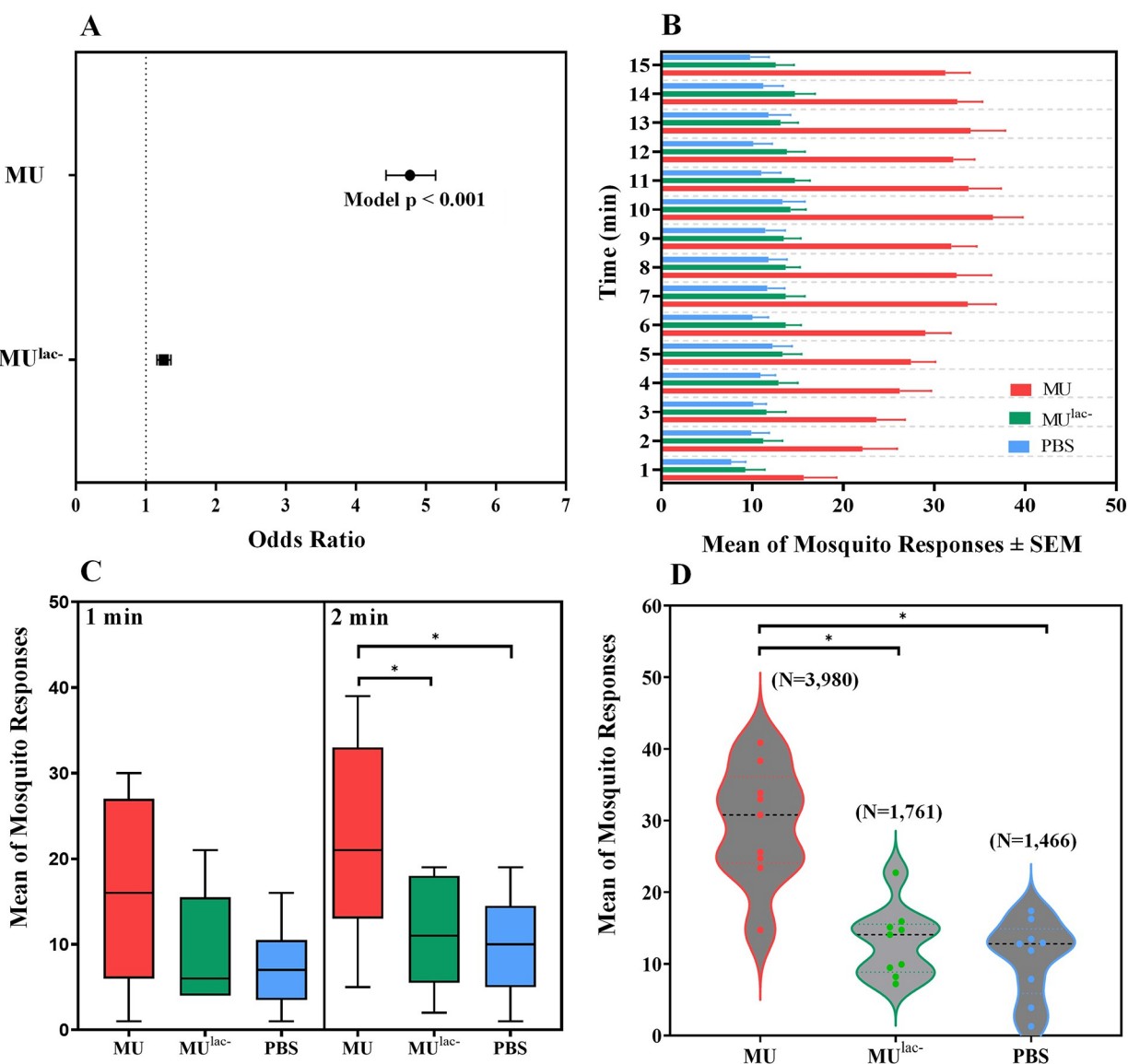

**Fig 2.** (A) The odds ratios of 3–5 d-old female *Ae. aegypti* mosquito responses to blood-feeders treated with MU (*M. ulcerans*) wildtype: mycolactone active, MU-mutant (MU^lac-^): mycolactone inactive, versus PBS as a control placed in parallel, 16.5 cm apart on the top of an 82 cm (L) x 45 cm (W) x 52 cm (H) Plexiglas cage during triplicate trials of 15 min with 50 mosquitos conducted at 24°C and 65% RH. (B) The mean number of *Ae. aegypti* mosquito responses per min ± SEM to blood-feeders treated with MU, MU^lac-^, and PBS as a control. (C) Box plots of *Ae. aegypti* mosquito responses during the initial 1 and 2 min of the trials (Black line, median; bounds of boxes, first and third quantiles; bars, range). (D) Violin plots highlight the distribution of *Ae. aegypti* mosquito responses for 15 min, with dots representing single trial and horizontal lines representing medians. Asterisks indicate statistically significant differences (p<0.05).

## RT-qPCR assay of open-tube MU response to mosquito proximity

Mycolactone upregulation was 24-fold (p = 0.006) greater for MU exposed to *Ae. aegypti* in Direct proximity (0.025 m) compared to MU cells with Far/None distance proximity (30.5 m). Those in Near proximity (1.83 m) did not show statistically different regulation from the Far/None (30.5 m) distance (Regulation = 0.175, p = 0.125) (Fig 3A).

**Table 1. Relative abundance of compounds ± SEM identified using GC-MS emitted from *M. ulcerans* 1615 (MU) and TN118 (MU[lac-]) with PBS as a control (CONT) from nonuplicate trials of quadruplicate replicates (n = 36) at 24 ± 0.5°C with 65 ± 5.0% RH; ± standard deviations.**

| Compound | Relative Abundance* (Mean±SEM) | | | | Retention Time (min) | Class |
|---|---|---|---|---|---|---|
| | MU[a] | PBS CTRL (MU)[b] | MU[lac- c] | PBS CTRL (MU[lac-]) [d] | | |
| 1Butanol,3-methyl | - | - | 7.134639 ±1.345685 | - | 4.17 | Fatty alcohols |
| Cyclopentanone | - | - | 0.356275 ±0.073263 | 0.262956±0.106315 | 5.93 | Hydrocarbons, Alicyclic |
| Octane (standard) | 1.000000 ±0.000000 | 1.000000 ±0.000000 | 1.000000 ±0.000000 | 1.000000±0.000000 | 6.2 | Hydrocarbon |
| 2-propanol,1-propoxy | - | 0.220356 ±0.112968 | - | - | 7.72 | Alcohols |
| p-Xylene | 0.158321 ±0.065142 | 0.389682 ±0.092802 | 0.040011 ±0.028588 | - | 8.96 | Hydrocarbons, Cyclic |
| Styrene | 0.255419 ±0.085857 | - | - | - | 9.72 | Hydrocarbons, Cyclic |
| Phenylamino | 0.084793 ±0.068302 | - | - | - | 12.29 | Aromatic amine |
| Benzaldehyde | 0.243813 ±0.031952 | 0.358921 ±0.045585 | 0.693219 ±0.083381 | 0.319193±0.117240 | 12.41 | Benzenoids Alcohols Ketones Aldehydes |
| Phenol | 0.300707 ±0.065601 | 0.418178 ±0.072086 | 0.328656 ±0.109580 | 0.302102±0.048804 | 12.87 | Alcohols Benzenoids |
| Benzene,1,2-dichloro | - | - | 0.311840 ±0.082992 | 0.358421±0.083712 | 14.35 | Hydrocarbons, Acyclic |
| Limonene | - | - | 0.430963 ±0.120389 | 0.558219±0.080538 | 14.85 | Terpenes |
| Acetophenone | 0.279270 ±0.128538 | 0.441175 ±0.133577 | - | 0.247361±0.124559 | 16.07 | Benzenoids Ketones |
| Nonanal | 0.475159 ±0.111224 | 0.547853 ±0.055003 | 0.456934 ±0.125135 | 0.218661±0.095765 | 17.34 | Aldehydes |
| (Trimethylsilyl) methyllithium | 0.082130 ±0.057790 | - | - | - | 20.05 | Carbocyclic |
| Benzothiazole | 2.396303 ±0.150229 | 2.335839 ±0.261206 | 3.907554 ±0.447900 | 5.845016±0.467230 | 21.25 | Benzenoids Thiazole Sulfur compound |
| Docosane,7-hexyl | 0.103465 ±0.070593 | 0.035740 ±0.035740 | - | - | 23.26 | Hydrocarbons, Acyclic |
| Nonadecane | 0.055036 ±0.055036 | 0.050770 ±0.050770 | 0.337891 ±0.140411 | 0.216313±0.132731 | 25.96 | Hydrocarbons, Acyclic |
| Dotriacontane | 0.104189 ±0.053472 | 0.014054 ±0.014054 | - | - | 25.97 | Hydrocarbons, Acyclic |
| Tetradecane, 2,6,10-trimethyl | 0.113055 ±0.061752 | 0.364409 ±0.097536 | 0.287683 ±0.116926 | 0.140115±0.093303 | 25.98 | Carbocyclic |
| Octadecanoic acid | - | - | - | 0.031663±0.021277 | 25.98 | Fatty alcohols |
| Tetracosane | 0.076757 ±0.051356 | - | 0.084746 ±0.069076 | - | 25.99 | Hydrocarbon |
| Naphthalene | 0.131660 ±0.079731 | - | - | - | 26.7 | Hydrocarbons, Cyclic |
| Quinoline | - | - | 0.021713 ±0.021713 | 0.061802±0.024717 | 27.18 | Heterocyclic |
| Eicosane | 0.095973 ±0.039608 | 0.284335 ±0.096686 | 0.214469 ±0.102427 | 0.394881±0.126974 | 28.54 | Hydrocarbons, Acyclic |
| Heneicosane | 0.040150 ±0.040150 | - | - | - | 28.55 | Alkanes |
| Butylated Hydroxytoluene | 1.024833 ±0.250773 | 0.977381 ±0.287978 | 0.380842 ±0.085848 | 0.590604±0.084411 | 28.68 | Benzenoids Alcohols |

*(Continued)*

**Table 1.** (Continued)

| Compound | Relative Abundance* (Mean±SEM) | | | | Retention Time (min) | Class |
|---|---|---|---|---|---|---|
| | MU[a] | PBS CTRL (MU)[b] | MU[lac- c] | PBS CTRL (MU[lac-]) [d] | | |
| Diethyl Phthalate | 0.500898 ±0.128446 | 0.439510 ±0.102345 | 0.220628 ±0.091201 | 0.342195±0.074162 | 30.71 | Carboxylic Acids |
| Methyl stearate | 0.177176 ±0.092885 | - | - | - | 33.26 | Fatty acid esters |
| 3beta-3-Lupanol | - | 0.072602 ±0.072602 | - | 0.015029±0.015029 | 45.46 | NA |

* The most abundant octane as an internal standard is assigned 1 and the others assigned a fractional percent of that value. a) MU 1615: *Mycobacterium ulcerans* (mycolactone active) b) MU 1615 CONT: MiddleBrook Agar (7H10) c) MU[lac-]: *Mycobacterium ulcerans* (mycolactone inactive) d) MU[l]

[ac-] CONT: MiddleBrook Agar (7H10) with Hygromycine (Final concentration 50 μg/ mL).

## MU versus MU[lac-] chromosome response to mosquito proximity

A PCA plot of normalized transcriptome data of each strain showed that MU[lac-] samples (designated as MU[lac-], Fig 3B) clustered together, and more so with MU with no mosquito contact (designated as None). Ten genes were significantly induced in MU compared to MU[lac-] due to

**Table 2. Volatile organic compounds uniquely expressed by *M. ulcerans* 1615 (MU) and by TN::118 (MU[lac-]) based on the relative frequency and abundance, from nonuplicate trials of quadruplicate replicates (n = 36) at 24 ± 0.5˚C with 65 ± 5.0% RH; ± standard deviations.**

| Compounds | Retention Time (min) | Relative Abundance* (Mean ± SEM) | Quantity Range (%) | Class | Reference |
|---|---|---|---|---|---|
| **UNIQUE TO MU:** | | | | | |
| Styrene | 9.72 | 0.255419±0.085857 | 87–97 | Hydrocarbons, Cyclic | 1) Bioconversion by Mycobacterium spp. M156 [56]<br>2) Genotoxic intermediate [57]<br>3) Human odor component [58]<br>4) Toxin to cells at higher concentrations [59]<br>5) Reproductive toxicants, neurotoxicants, or carcinogens in vivo or vitro [60]<br>6) Functional equivalency in quorum sensing in Candida albicans [61] |
| Phenylamino | 12.29 | 0.084793±0.068302 | 90–94 | Aromatic amine | 1) Degradation of aminoacylation activity in M. tuberculosis [62]<br>2) Association with autoinducer (AI) -3 [63] |
| (Trimethylsilyl) methyllithium | 20.05 | 0.082129±0.057790 | 83 | Carbocyclic | 1) Intramolecular coordination [64] |
| Naphthalene | 26.7 | 0.131659±0.079731 | 81–90 | Hydrocarbons, Cyclic | 1) Human odor component [65]<br>2) Mosquito attraction [66]<br>3) Cattle odor component [67] |
| Heneicosane | 28.55 | 0.040149±0.040150 | 93–96 | Alkanes | 1) Attraction for oviposition [68, 69]<br>2) Metabolite in mycobacterium spp. [70] |
| Methyl stearate | 33.26 | 0.177176±0.092885 | 97 | Fatty acid esters | 1) Metabolite in Mycobacterium smegmatis [71]<br>2) Main compound in extracellular hydrophobic metabolite [72] |
| **UNIQUE TO MU[lac-]:** | | | | | |
| 1Butanol,3-methyl | 4.17 | 7.134639±1.345685 | 86–94 | Fatty alcohols | 1) Fragrance and flavoring in industry [73]<br>2) A chemical in the pheromone used by insects [74–76]<br>3) Carrion odor of dead beetles [77] |

*The most abundant octane as an internal standard is assigned 1 and the others assigned a fractional percent of that value.

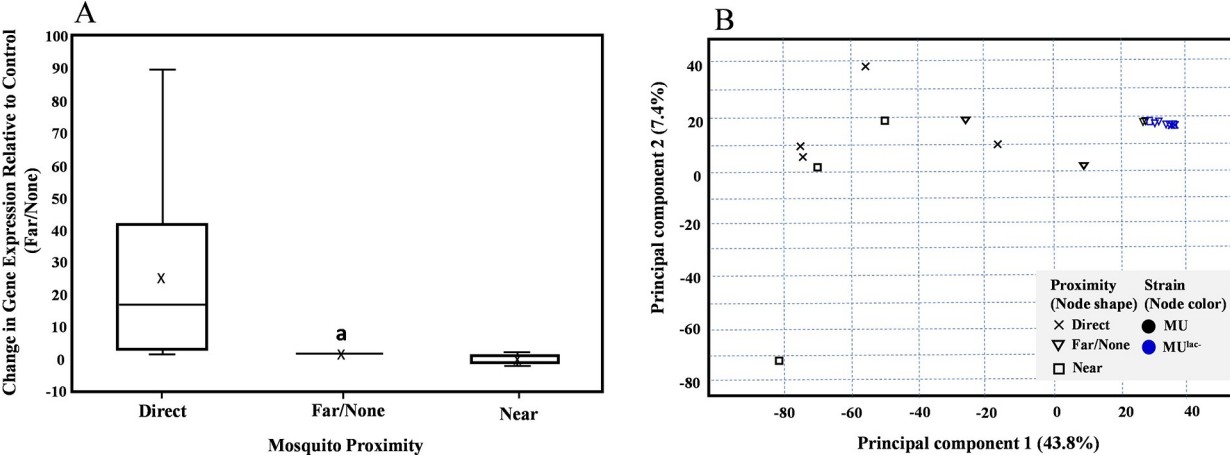

**Fig 3.** (A) Relative Normalized Expression of the Enoyl Reductase (ER) Gene. Results of relative normalized expression of the ER gene following exposure to newly emerged adult mosquitoes for 24 h at 3 different distances in the same room or separated by a different room. The *ppk* gene was used as a housekeeping gene for normalization. [a]Significance difference between MU in Direct mosquito proximity compared to MU in Far/None proximity (p = 0.006). (B) Principal component analysis (PCA) of transcriptome data clustering gene expression according to *M. ulcerans* strain type and proximity to mosquitoes. Strains are differentiated by color. MU: *M. ulcerans* wildtype (mycolactone active) are shaded black; MU[lac-]: *M. ulcerans* mutant (mycolactone inactive) are shaded blue. Proximity to mosquitoes is differentiated by shape. Direct: Direct proximity to mosquitoes with nodes shaped as x; Far/None: Far/no proximity to mosquitoes with nodes shaped as ▽; Near: Near proximity to mosquitoes with nodes shaped as ☐.

mosquito proximity (Table 3). In a comparison of MU versus MU[lac-], all ten genes showed significantly modulated expression whether having direct access to or in near mosquito proximity, with the exception of MU[lac-] in near mosquito proximity versus MU in near mosquito proximity for the *ssr* gene (p = 0.247). There was only one statistically significant difference in MU compared to MU[lac-] in the absence of mosquito contact (None), MUL_RS17465, encoding SigB (p = 0.026).

No statistically significant differences were found among genes when comparing according to MU[lac-] proximity to mosquitoes. However, four genes present in MU[lac-] None were not expressed (or was below detection levels) upon mosquito direct contact (MUL_RS15540, *clpC*1, *lon*, and MUL_RS17465) and three genes were not expressed (or were below detection levels) upon near contact (MUL_RS05050, MUL_RS15540, and MUL_RS16005). Only one gene present in MU[lac-] None, was not expressed (or were below detection levels) in either near or direct contact (MUL_RS15540) and one gene not present in MU[lac-] None, was expressed in both near and direct contact (MUL_RS25225).

Expressed genes were not significantly different between MU Direct and MU Near to mosquitoes. However, statistical differences in expressed genes were found between MU in near mosquito proximity and those in the absence of mosquito contact (MU None) for MUL_RS15460 (p = 0.007) and MUL_RS15540 (p = 0.015), and between MU in Direct mosquito contact versus MU in the absence of mosquito contact (None) for *clpB* (p = 0.010), MUL_RS15540 (p = 0.007), *clp*C1 (p = 0.037), *lon* (p<0.001), MUL_RS17465 (p = 0.004), and MUL_RS25225 (p = 0.010).

## MU pMUM001 response to mosquito proximity

Ten genes in the pMUM001 plasmid were significantly induced in MU, according to proximity with mosquitoes (Table 4). The induced genes encoded RepA and ParA (genes thought to be part of a regulatory cluster, MUL_RS00095, and MUL_RS00090) a putative insertion sequence, a DEAD/DEAH helicase (MUL_RS00335), a type I polyketide synthase (MUL_RS00210), and two additional conserved hypothetical proteins (MUL_RS27465 and

**Table 3. Statistically significantly different mean read counts of normalized *M. ulcerans* 1615 (MU) and TN::118 (MU^lac-^) mapping to *M. ulcerans* chromosome according to mosquito proximity: Direct, Near (3mm), and None.** Transcripts are listed as gene names or locus tags according to GenomeNet RefSeq database for NC_008611 (https://www.genome.jp/).

| Name | Encoded Protein | Log2 Fold Change | FDR p-value | Mean±SD | | | | | |
|---|---|---|---|---|---|---|---|---|---|
| | | | | MU^lac-^ | MU | MU^lac-^ | MU | MU^lac-^ | MU |
| | | | | None | None | Near | Near | Direct | Direct |
| MUL_RS05050 | DHA2 Efflux MFS Transporter Permease | 5.79 | 0 | 116.04 ±232.09 | 7810.94 ±12159.02 | 0±0 | 17579.49 ±2182.93† | 538.22 ±1076.44 | 18662.43 ±4668.96‡ |
| MUL_RS15640 | alkyl hydroperoxide reductase | 3.33 | 0 | 511.66 ±468.72 | 4107.35 ±5148.54 | 202.45 ±350.65 | 22270.35 ±12830.18† | 1323.85 ±1581.13 | 16472.81 ±4304.32‡ |
| clpB | ATP dependent chaperone clpB | 3.08 | 0 | 797.88 ±485.85 | 6063.09 ±8131.24 | 121.47 ±210.39 | 16416.51 ±1231.05† | 2152.87 ±4305.75 | 19853.74 ±5360.57‡ |
| ssrA | maleylpyruvate isomerase family mycothiol-dependent enzyme | 2.59 | 0 | 8674.82 ±2794.46 | 40300.56 ±46448.24 | 17509.04 ±22131.58 | 69809.65 ±14545.69† | 1614.66 ±3229.31 | 85006.93 ±20057.37‡ |
| MUL_RS15540 | Cytochrome P450 | 6.13 | 0 | 38.68 ±77.36 | 1345.63 ±1633.40 | 0±0 | 5899.46 ±2770.76† | 0±0 | 5903.50 ±1974.79‡ |
| clpC1 | ATP-dependent protease ATP-binding subunit CLpC | 4.41 | 0.01 | 78.82 ±91.05 | 1455.21 ±1646.41 | 40.49±70.13 | 2888.82 ±429.97† | 0±0 | 3442.27 ±640.03‡ |
| lon | Endopeptidase La | 3.55 | 0.02 | 96.74 ±116.11 | 943.82 ±1166.07 | 80.98±140.26 | 2304.38 ±112.34† | 0±0 | 3589.05 ±652.21‡ |
| MUL_RS17465 | Sigma 70 family RNA polymerase sigma factor | 3.9 | 0.02 | 58.06 ±116.13 | 757.25 ±547.70* | 40.49±70.13 | 1198.73 ±182.74† | 0±0 | 1627.28 ±175.75‡ |
| MUL_RS16005 | PPE family protein | 3.94 | 0.03 | 116.09 ±134.04 | 2031.9 ±2691.66 | 0±0 | 3638.86 ±82.72† | 538.22 ±1076.44 | 4517.41 ±708.76‡ |
| MUL_RS25225 | Rieske 2Fe-2S domain Containing Protein | 4.36 | 0.04 | 0±0 | 834.34 ±998.64 | 60.73±105.20 | 3488.10 ±1265.16† | 538.22 ±1076.44 | 5386.98 ±3050.51‡ |

*Significant gene induction in MU versus MU^lac-^ without having any contact with mosquitoes.

†Significant gene induction in MU versus MU^lac-^ when in near proximity to mosquitoes.

‡ Significant gene induction in MU versus MU^lac-^ when directly having contact with mosquitoes.

**Table 4. Statistically significant mean read counts mapping to *M. ulcerans* pMUM001 according to mosquito proximity: Direct, Near (3mm), and None.** Log2 fold change is in relation to *M. ulcerans* 1615 (MU) with no mosquito contact. Transcripts are listed as gene names or locus tags according to GenomeNet RefSeq database for NC_008611 (https://www.genome.jp/).

| Name | Encoded Protein | Log$_2$Fold Change | FDR p-value | Mean±SD | | |
|---|---|---|---|---|---|---|
| | | | | MU None | MU Near | MU Direct |
| MUL_RS00005 | RepA | -66.57 | 0.023 | 0±0 | 3478.67±2199.71a | 2578.84±1365.02 |
| MUL_RS00025 | ParA family Protein | -64.61 | 0.014 | 0±0 | 2616.54±1854.39 | 3476.39±1504.04[a] |
| MUL_RS00090 | FHA domain containing protein | -3.31 | 0.002 | 4394.06±5130.35 | 14433.19±7655.64 | 24503.69±3020.41[a] |
| MUL_RS00095 | Sensor domain containing Protein | -107.71 | 0.012 | 0±0 | 5645.08±1326.77[a] | 4478.31±3119.32[a] |
| MUL_RS00105 | WhiB family transcriptional regulator | -86.8 | 0.008 | 0±0 | 4690.52±2219.75[a, c] | 1012.91±1540.87b |
| MUL_RS00150 | transposase | -3.82 | 0.035 | 5586.06±4846.00 | 13133.06±4603.08 | 22013.63±10025.51[a] |
| MUL_RS00210 | Type I Polyketide Synthase | -2.93 | 0.007 | 17495.4±11973.36 | 52682.70±57177.34a | 43236.30±11922.32[a] |
| MUL_RS00335 | Dead/DEAH box helicase | -2.15 | 0.019 | 1735.4±3470.81 | 4186.09±1222.79 | 8489.17±2289.46[a] |
| MUL_RS27465 | hypothetical protein | -7.23 | 0.012 | 347.08±694.16 | 2220.64±1172.28a, c | 116.23±232.46b |
| MUL_RS00405 | hypothetical protein | -54.67 | <0.001 | 0±0 | 2686.89±936.35[a] | 2911.61±572.08a |

[a]Significant gene expression in MU treatment compared to MU None.

[b]Significant gene expression in MU treatment compared to MU Near.

[c]Significant gene expression in MU treatment compared to MU Direct.

MUL_RS00405). Five of the ten genes (MUL_RS00005, MUL_RS00025, MUL_RS00095, MUL_RS00105, and MUL_RS00405) were newly induced by Near proximity; the remaining five were already expressed but increased in expression. Significant differences were found between MU in Near mosquito proximity (MU Near) compared to MU None for all of these genes except for MUL_RS00150 (p = 0.621), MUL_RS00090 (p = 0.113), MUL_RS00335 (p = 0.766), and MUL_RS00025 (p = 0.092).

Nine of the ten genes were induced with direct mosquito contact (MU Direct), compared to no mosquito contact (MU None). The same five genes as were induced by near proximity, were newly induced by direct contact (MUL_RS00005, MUL_RS00025, MUL_RS00095, MUL_RS00105, and MUL_RS00405); and of the remaining five that were already expressed, all increased in expression except MUL_RS27465, a hypothetical protein, which decreased in expression. Significant differences were found for all genes when comparing MU Direct with MU None except for MUL_RS0005 (p = 0.089), MUL_RS00105 (p = 1.00), and MUL_RS27465 (p = 1.00). There were no significant differences between transcript counts from MU in direct mosquito proximity and MU in near mosquito proximity, except for MUL_RS27465 (p = 0.017) and MUL_RS00105 (p = 0.032).

## Discussion

To our knowledge, this study provides the first evidence of interkingdom interactions between viable MU and *Ae. aegypti*, and its possible amplification through the secondary metabolite mycolactone. Mycolactone functions as a cue, or potentially an interkingdom signal, for mosquitoes, which could lead to host allocation or oviposition sites as demonstrated in previous research [40, 43]. Mycolactone has unique molecular properties [40], suggesting its (or its degradation products) potential to produce candidate VOCs affecting mosquito behavior as related to host-seeking or oviposition. Mosquitoes are primarily guided to locate a suitable host by VOCs produced by bacteria; for example, *Staphylococcus epidermidis*, a common bacterium of the human skin flora, is a biological mediator of mosquito attraction and blood-feeding [78, 79]. Verhulst et al. [80] demonstrated *S. epidermidis* on blood agar plates were more attractive than sterile blood agar plates to *Anopheles gambiae* (Diptera: Culicidae) *sensu stricto*, a vector for the malaria parasite. These results likely explain why washing feet with anti-bacterial soap results in *An. gambiae* shifting blood feeding to other body parts [81], indicates an interaction between the mosquito and the human skin microbiota.

In this study, we determined *Ae. aegypti* attraction to blood-feeders treated with MU resulted in 126% greater attraction to the blood-feeder when compared with MU[lac-]. Seven VOC compounds, not found in the diluent PBS, were differentially expressed between MU and MU[lac—](Table 2). The six compounds (styrene, phenylamino, (trimethylsilyl) methyl-lithium, naphthalene, heneicosane, and methyl stearate) present in MU, but not in MU[lac-], were associated with a wide range of biological importance ranging from QS to human odor components and could contribute to the differential behavioral responses measured in this study. There was one compound present in MU[lac-], but not in MU, fragrant primary alcohol, 1Butanol,3-methyl (also known as isoamyl alcohol). This compound is used in the fragrance and flavoring industry, is a component of some insect pheromones, and is found during the decomposition of insects. Results suggest that it should be explored as a possible repellent cue.

At the RNA level, MU showed significant induction of ten chromosomal genes compared to MU[lac-]. Several of these, including those encoding for ClpC1, Lon protease, ClpB, SigB, and alkyl hydroperoxide reductase are involved in stress responses [82]. MUL_RS16005 encodes a PPE 30 membrane protein that may also be controlled by SigB and may be induced as an adaptation to environmental stimuli associated with mosquito proximity. PE/PPE protein

induction for environmental adaptation has been shown in *M. marinum*, where sigma factors, along with WhiB4, regulate PE/PPE gene families and are essential for virulence [83, 84]. Additionally, SsrA, involved in protein tagging, directed degradation (in association with ClpC1), and ribosome rescue, is also a stress response and has been induced in other mycobacterial systems in response to ribosome inhibiting antimicrobial agents [85]. Another induced gene, MUL_RS05050 encodes a DHA2 efflux MFS transporter permease that is an integral membrane transport protein with potential functions ranging from multidrug efflux (such as in *M. tuberculosis*), to transport of bacterial metabolites, QS molecules, and virulence factors [86]. Finally, MUL_RS15540 encodes a cytochrome p450 with an unknown function. Cytochrome p450s are plentiful in many slow-growing mycobacteria and possess multiple important biochemical functions, such as lipid metabolism, secondary metabolite production, and pathogenicity [87, 88].

Significant differential pMUM001 gene expression of MU was also observed associated with the proximity of mosquito contact in this study. Ten pMUM001 genes were significantly induced in response to mosquito proximity. Two are associated with plasmid replication, including the predicted product of *rep*A and *par*A, a gene encoding a chromosome partitioning protein, required for plasmid segregation upon cell division [89]. Also induced were genes thought to be part of a regulatory gene cluster including a gene encoding a protein of unknown function (MUL_RS00090) but containing a phosphopeptide recognition domain (possibly promoting phosphorylation-dependent protein-protein interactions [90], a probable conserved membrane protein (MUL_RS0095), and a WhiB-like transcriptional regulator (MUL_RS00105). Of note, MUL_RS00210, encoding *mls*A2, required in part, for the synthesis of the mycolactone core was also induced, as were two conserved hypothetical proteins of unknown function, and one encoding a DEAD/DEAH helicase, an enzyme essential in RNA metabolism and signaling/gene regulation [91]. However, other genes necessary for mycolactone syntheses, such as *mls*A1 and *mls*B, were not differentially expressed according to proximity. One reason for this could be due to the short mosquito exposure time (15 minutes); however, this would require further experimentation to determine. We also conducted a longer-term experiment where we exposed opened tubes of $10^6$ CFU/mL MU to adult *Ae. aegypti* at Direct, Near, or Far/None proximity for 24 h, and measured gene expression of the ER gene [12]. Mean data showed ER upregulation 24-fold (p = 0.006) from MU directly exposed to *Ae. aegypti* compared to MU cells at near distance proximity (1.83 m), which also was not statistically different from the far (30.5 m) distance (Fig 3A).

Altogether, the finding that strain drove significantly different gene induction with comparative proximity of mosquitoes in MU versus MU$^{lac-}$ suggests that acquisition of the plasmid conferred a more sensitive stress response to mosquito proximity in MU. Whether this is in response to mosquito VOCs or mosquito microbiome VOCs is unknown. Interestingly, there was only one gene showing a significant difference between MU and MU$^{lac-}$ in the absence of mosquitoes, that encoding SigB (MUL_RS17465), suggesting a broader role for SigB in the absence of mosquitoes. In general, SigB functions in stress response and bacterial survival, such as in *Mycobacteria*, under many adverse environmental conditions [92, 93]; but can affect logarithmic growth in cells, which would be beneficial for MU environmental persistence.

Overall, these data suggest that mycolactone production may, in part, be a response to mosquito signals and proximity, or to mosquito microbiome signals and proximity. This exciting finding suggests that mosquito-microbiome-MU communication can regulate MU expression as well as mosquito behavior, a finding with implications for MU control, but also more broadly in understanding disease ecology, pathogen dispersal, and vector attraction. However, it is important to note that these data only reflect gene expression and should be repeated to also include measuring mycolactone production. And additional in-depth research linking

mycolactone production (i.e., chemical complementation of MU$^{lac-}$) with VOC profiles generated, and corresponding mosquito blood-feeding is necessary to further elucidate mycolactone's ecological relevance in this environmental context.

Bacteria interact with each other through specific communications pathways (i.e., autoinducer in QS). Such responses to these compounds are tightly linked with bacterial density [94]. In fact, QS from bacteria elicits host (e.g., humans, plants, other multicellular organisms) responses [95]. Mycolactone could function as a QS antagonist [96] or regulator of biological activities, such as symbiosis [97], virulence [98, 99], or conjugation [100]. Such ability could allow MU colonization and persistence within a given environment (e.g., water or human skin) [101].

The ability of bacteria to QS has been demonstrated to regulate mosquito attraction to a blood-meal. Zhang et al. [47] determined mosquitoes are more attracted (74.0%) to wildtype *S. epidermidis* (i.e., commensal on human skin; able to QS) than accessory gene regulator (*agr*) mutant *S. epidermidis* (unable to QS). Similar to this study, adult blow fly *Lucilia sericata* (Meigen) (Diptera: Calliphoridae) response to *P. mirabilis*, a bacterium associated with carrion (i.e., source for larval development), was partly tied to QS [102]. This evidence suggested there may be widespread interkingdom interactions between mosquitoes and microbes that have biological and ecological importance within an environment.

While results were consistent across experiments, limitations of the current study were identified. Our approach when examining MU-mosquito interactions was to use a single bacterial species and determine its impact on the mosquito of interest. Such an approach is known to be limiting in terms of deciphering the true ecological relevance of bacterial interactions with mosquitoes, as the bacterial activity in isolation can be quite different than in the community mixtures typically encountered in a complex and dynamic ecosystem (e.g., human skin). Furthermore, morphological (Fig 1A and 1B) and physiological differences between MU and MU$^{lac-}$ may have an influence on mosquito behavior, particularly host-seeking behavior; mycolactone is UV active and MU$^{lac-}$ lacks pigmentation and is not UV active [103]. The role of color vision in mosquito host-seeking behavior has been ascertained with the exception of a few studies [104, 105], showing *Ae. aegypti* had relatively poor acuity but is capable of specific wavelength discrimination (323 nm ~ 621 nm) [32]. To determine their impact on bacteria-mosquito interactions, and reduce the variability, factors, such as vision, should be examined in greater detail. Furthermore, our research explored MU-mosquito interactions under set conditions. And it is known olfactory thresholds of mosquito response are significantly influenced by environmental factors such as temperature and humidity [106]. For example, the activity of antennal receptor neurons on *Ae. aegypti* tended to be optimal at 26 to 28˚C [34] and humidity has been reported as a factor to regulate mosquito attraction [107]. Also, the individual compound in response to ecological characteristics that affect mosquito behavior should be examined in greater detail. These predictions are amenable to testing in the lab and applicable in a field setting.

## Conclusions

In summary, a triple-choice assay and GC-MS analysis provide clear evidence that mycolactone serves as an interkingdom cue with mosquitoes. Also, we demonstrate through gene expression analysis significant differential gene expression of MU versus MU$^{lac-}$ in both contact and near proximity to *Ae. aegypti*. Results suggest that the relationship between the pathogen and mosquito could be potentially more closely evolved than previously thought (i.e., mycolactone may be a signal rather than a cue) and the possibility of developing novel methods for shifting VOC profiles by manipulating the mycolactone system of host-associated

invasive bacteria, resulting in reduced mosquito attraction and oviposition. A recent study showed specific bacteria-associated VOCs (e.g., indole and skatole) function as both possible host-seeking attractants and oviposition stimulants for mosquitoes [108–110]. Additionally, our group has previously demonstrated the importance of interkingdom communication between insects and bacteria in response to host-seeking and oviposition site selection behavior based on QS pathways [47, 102]. This, and our previous studies, point to the potential of mycolactone association with shifting VOC production and should be further explored for the development of new and effective methods for mosquito control and suppression of pathogen transmission. On a broader scale, our work highlights the relevance of interkingdom interactions for structuring disease ecology dynamics.

## Supporting information

**S1 Fig. Structure of mycolactone A/B, its known molecular targets, and cellular effects.** Original mycolactone figure obtained by Lrandolp—Own work, CC BY-SA 3.0, https:// commons.wikimedia.org/w/index.php?curid=7007015.
(TIF)

**S2 Fig. Principal component analysis (PCA) was conducted on the volatile organic compounds (VOCs) emitted from *M. ulcerans* 1615 (MU) and TN118 (MU$^{lac-}$) with PBS serving as the control.** (A) Loading plot displays the relationships and (B) score plot shows the distribution of the samples based on their scores in PC1 and PC2. Replicate samples are represented by dots of the same color.
(TIF)

**S3 Fig. (A) Relative abundance of the volatile organic compounds (VOCs) emitted from *M. ulcerans* 1615 (MU) and TN118 (MU$^{lac-}$) with PBS serving as the control.** Asterisk (*) and pound sign (#) represent VOCs uniquely expressed by MU or MU$^{lac-}$, respectively. (B) Bar chart of internal standard ratioed peak area values (Mean ± SEM) for seven selected VOCs that were unique to MU or MU$^{lac-}$ samples. The octane as an internal standard (blue dashed line) is assigned 1 and the others assigned a fractional percent of that value.
(TIF)

## Acknowledgments

The authors thank Dr. Jason Rosch, Department of Infectious Diseases at St. Jude Children's Research Hospital, for help with the RNASeq analyses; Dr. Johnathan Cammack for their help with volatile analysis; Dr. Steve Sweet and the Geochemical and Environmental Research Group (GERG) at Texas A&M University for assistance with GC-MS sample processing.

## Author Contributions

**Conceptualization:** Tawni L. Crippen, Jeffery K. Tomberlin, Heather R. Jordan.

**Data curation:** Dongmin Kim, Tawni L. Crippen, Laxmi Dhungel, Heather R. Jordan.

**Formal analysis:** Dongmin Kim, Tawni L. Crippen, Laxmi Dhungel, Heather R. Jordan.

**Funding acquisition:** Tawni L. Crippen, Jeffery K. Tomberlin.

**Investigation:** Heather R. Jordan.

**Methodology:** Dongmin Kim, Tawni L. Crippen, Laxmi Dhungel, Pablo J. Delclos, Jeffery K. Tomberlin, Heather R. Jordan.

**Project administration:** Tawni L. Crippen.

**Resources:** Tawni L. Crippen, Jeffery K. Tomberlin.

**Supervision:** Tawni L. Crippen, Pablo J. Delclos, Jeffery K. Tomberlin, Heather R. Jordan.

**Validation:** Dongmin Kim, Tawni L. Crippen, Jeffery K. Tomberlin, Heather R. Jordan.

**Visualization:** Dongmin Kim, Tawni L. Crippen, Heather R. Jordan.

**Writing – original draft:** Dongmin Kim, Tawni L. Crippen, Laxmi Dhungel, Jeffery K. Tomberlin, Heather R. Jordan.

**Writing – review & editing:** Dongmin Kim, Tawni L. Crippen, Jeffery K. Tomberlin, Heather R. Jordan.

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
