## [Decision Letter · Decision Letter 0]

13 Jun 2023

PONE-D-23-13191Behavioral interplay between mosquito and mycolactone produced by Mycobacterium ulcerans and bacterial gene expression induced by mosquito proximityPLOS ONE

Dear Dr. Kim,

Thank you for submitting your manuscript to PLOS ONE. After careful consideration, we feel that it has merit but does not fully meet PLOS ONE’s publication criteria as it currently stands. Therefore, we invite you to submit a revised version of the manuscript that addresses the points raised during the review process.

We look forward to receiving your revised manuscript.

Kind regards,

Mozaniel Santana de Oliveira, Ph.D

Academic Editor

PLOS ONE

Journal Requirements:

"NO authors have competing interests" 

Reviewers' comments:

Reviewer's Responses to Questions

**Comments to the Author**

1. Is the manuscript technically sound, and do the data support the conclusions?

Reviewer #1: Yes

Reviewer #2: Partly

2. Has the statistical analysis been performed appropriately and rigorously? 

Reviewer #1: Yes

Reviewer #2: Yes

3. Have the authors made all data underlying the findings in their manuscript fully available?

Reviewer #1: Yes

Reviewer #2: No

4. Is the manuscript presented in an intelligible fashion and written in standard English?

Reviewer #1: Yes

Reviewer #2: Yes

5. Review Comments to the Author

Reviewer #1: MS entitled, ‘Behavioral interplay between mosquito and mycolactone produced by Mycobacterium ulcerans and bacterial gene expression induced by mosquito proximity’ by Dongmin Kim is really interesting piece for me. I find it very enjoying while reviewing this manuscript. Below are my few queries to be addressed by authors:

1. In introduction, I suggest authors to add more about the mycolactone. Is there any chemical structure available? If yes, please add it. Further, please add its biosynthetic pathway by which it could be produced.

2. Moreover, authors need to add the schematic diagram explaining plausible molecular targets.

3. ‘ ml’ should be written as ‘mL’ throughout the manuscript.

4. Figure 1(B) needs to be replaced with more additional clarified image.

5. Authors can supplement GC-MS results as a supporting material. Or provide for review.

Reviewer #2: The manuscript entitled "Behavioral interplay between mosquito and mycolactone produced by Mycobacterium ulcerans and bacterial gene expression induced by mosquito proximity" is well organized and brings important advances in its field of research. However, some improvements need to be made to make it suitable for publication in this journal.

1) The experimental section is very complex, with many details that make the reading tiresome and confusing. I suggest that the author be more succinct in his writing, simplifying and making the text more attractive.

2) The presence of diethyl phthalate (Table 1) looks like plastic contamination because since the compound is synthetic, it is unlikely to appear naturally in the sample studied. If the authors have used any material used in the extraction of the volatiles made of plastic, I suggest that the identification of the compounds be reviewed.

3) The data provided in the manuscript are insufficient to determine the identity of the compounds. I suggest that the authors insert chromatograms and the mass spectra (for the majority compounds). This will facilitate the verification of the identity of the chemical compounds.

6. PLOS authors have the option to publish the peer review history of their article (what does this mean?). If published, this will include your full peer review and any attached files.

Reviewer #1: **Yes: **Suraj N. Mali

Reviewer #2: No

---

## [Author Response · Author response to Decision Letter 0]

6 Jul 2023

Review Comments to the Author

Reviewer #1: 

MS entitled, ‘Behavioral interplay between mosquito and mycolactone produced by Mycobacterium ulcerans and bacterial gene expression induced by mosquito proximity’ by Dongmin Kim is really interesting piece for me. I find it very enjoying while reviewing this manuscript. Below are my few queries to be addressed by authors:

The authors greatly appreciate the time and effort taken by the reviewer to improve our submission. Our responses are documented below:

1. In introduction, I suggest authors to add more about the mycolactone. Is there any chemical structure available? If yes, please add it. Further, please add its biosynthetic pathway by which it could be produced.

The authors agree and appreciate the reviewer’s insight. The mycolactone descriptions including chemical structure and biosynthesis pathway have been added to introduction (Line # 58).

2. Moreover, authors need to add the schematic diagram explaining plausible molecular targets.

Descriptions for the molecular targets have been incorporated into the introduction, specifically on Line 61. Additionally, we have included a schematic diagram related to this information, which has been uploaded as supporting information (S1 Fig).

3. ‘ml’ should be written as ‘mL’ throughout the manuscript.

The authors apologize for missing these obvious mistakes. Corrections have been made to the manuscript.

4. Figure 1(B) needs to be replaced with more additional clarified image.

The high-resolution image in Figure 1(B) has been updated.

5. Authors can supplement GC-MS results as a supporting material. Or provide for review.

GC-MS results including loading/ score plots, Composition ratio, and bar chart of internal standard ratioed peak area values have been uploaded as supporting information. 

Reviewer #2: 

The manuscript entitled "Behavioral interplay between mosquito and mycolactone produced by Mycobacterium ulcerans and bacterial gene expression induced by mosquito proximity" is well organized and brings important advances in its field of research. However, some improvements need to be made to make it suitable for publication in this journal.

The authors thank Reviewer #2 for your thoughtful criticism of our work. In response to these suggestions, we have made incorporated edits when possible and to improve clarity.

1) The experimental section is very complex, with many details that make the reading tiresome and confusing. I suggest that the author be more succinct in his writing, simplifying and making the text more attractive.

The authors agree and the experimental section has been revised.

2) The presence of diethyl phthalate (Table 1) looks like plastic contamination because since the compound is synthetic, it is unlikely to appear naturally in the sample studied. If the authors have used any material used in the extraction of the volatiles made of plastic, I suggest that the identification of the compounds be reviewed.

You have raised an important point here. The authors concur that diethyl phthalate, which has low volatility, is unlikely to have originated from the samples. It raises suspicion that this compound could have been collected from the parafilm used to seal the glass filtering jar during VOC collection. Table 1 provides evidence that nine compounds, including diethyl phthalate, were present in all treatments.

3) The data provided in the manuscript are insufficient to determine the identity of the compounds. I suggest that the authors insert chromatograms and the mass spectra (for the majority compounds). This will facilitate the verification of the identity of the chemical compounds.

We completely agree with your comment. GC-MS results including loading/ score plots, Composition ratio, and bar chart of internal standard ratioed peak area values have been uploaded as supporting materials. 

Finally, we look forward to responding to any further questions and comments you may have. We thank the reviewers for their time and effort again.

---

## [Decision Letter · Decision Letter 1]

26 Jul 2023

Behavioral interplay between mosquito and mycolactone produced by Mycobacterium ulcerans and bacterial gene expression induced by mosquito proximity

PONE-D-23-13191R1

Dear Dr. Kim,

We’re pleased to inform you that your manuscript has been judged scientifically suitable for publication and will be formally accepted for publication once it meets all outstanding technical requirements.

Kind regards,

Mozaniel Santana de Oliveira, Ph.D

Academic Editor

PLOS ONE

Additional Editor Comments (optional):

Reviewers' comments:

Reviewer's Responses to Questions

**Comments to the Author**

1. If the authors have adequately addressed your comments raised in a previous round of review and you feel that this manuscript is now acceptable for publication, you may indicate that here to bypass the “Comments to the Author” section, enter your conflict of interest statement in the “Confidential to Editor” section, and submit your "Accept" recommendation.

Reviewer #1: All comments have been addressed

2. Is the manuscript technically sound, and do the data support the conclusions?

Reviewer #1: Partly

3. Has the statistical analysis been performed appropriately and rigorously? 

Reviewer #1: Yes

4. Have the authors made all data underlying the findings in their manuscript fully available?

Reviewer #1: Yes

5. Is the manuscript presented in an intelligible fashion and written in standard English?

Reviewer #1: Yes

6. Review Comments to the Author

Reviewer #1: Accept. Authors have addressed my points. manuscript can now be accepted. its a good piece of article, now.

7. PLOS authors have the option to publish the peer review history of their article (what does this mean?). If published, this will include your full peer review and any attached files.

Reviewer #1: **Yes: **NA

---

## [Editor Report · Acceptance letter]

27 Jul 2023

PONE-D-23-13191R1 

Behavioral interplay between mosquito and mycolactone produced by *Mycobacterium ulcerans* and bacterial gene expression induced by mosquito proximity 

Dear Dr. Kim:

I'm pleased to inform you that your manuscript has been deemed suitable for publication in PLOS ONE. Congratulations! Your manuscript is now with our production department. 

Kind regards, 

on behalf of

Dr. Mozaniel Santana de Oliveira 

Academic Editor

PLOS ONE